# Protection of Galacto-Oligosaccharide against *E. coli* O157 Colonization through Enhancing Gut Barrier Function and Modulating Gut Microbiota

**DOI:** 10.3390/foods9111710

**Published:** 2020-11-21

**Authors:** Yan Zou, Jin Wang, Yuanyifei Wang, Bo Peng, Jingmin Liu, Bowei Zhang, Huan Lv, Shuo Wang

**Affiliations:** Tianjin Key Laboratory of Food Science and Health, School of Medicine, Nankai University, Tianjin 300071, China; 2120181361@mail.nankai.edu.cn (Y.Z.); wangjin@nankai.edu.cn (J.W.); wangyyf163@163.com (Y.W.); iq1226jsnpb@hotmail.com (B.P.); liujingmin@nankai.edu.cn (J.L.); bwzhang@nankai.edu.cn (B.Z.); lvhuan@nankai.edu.cn (H.L.)

**Keywords:** galacto-oligosaccharide, *Escherichia coli* O157:H7, reduce colonization, intestinal microflora

## Abstract

Galacto-oligosaccharide (GOS) has been added to infant formula as prebiotics and can bring many benefits to human health. This study proved the effect of GOS in prevention and alleviation against *E. coli* O157 invasion and colonization and the mechanism behind this was explored in a mice model. The results showed that the expression of Muc2 and Occlaudin were both significantly down-regulated (*p* < 0.05) by *E. coli* O157 infection, while GOS alleviated this phenomenon, which means that GOS can reduce the colonization of *E. coli* O157 by enhancing the gut barrier function. Through the determination of inflammatory cytokines, we found that GOS can relieve inflammation caused by pathogens. At the same time, GOS can promote the growth of probiotics such as *Akkermansia*, *Ruminococcaceae* and *Bacteroides*, thus modulating microorganism environments and improving short chain fatty acid (SCFA) levels in the intestine. This study provides an explanation for the mechanism behind the protection of GOS against pathogen infection.

## 1. Introduction

Enterohemorrhagic *Escherichia coli* O157:H7 (*E. coli* O157) is a leading food-borne pathogen of intestinal infection. It is commonly seen in meat, dairy and many other products, and can cause diarrhea, hemorrhagic colitis, and hemolytic uremic syndrome worldwide [1,2]. The pathogenic site of *E. coli* O157 is mainly the colon and ileum. Antibiotics are mainly used to deal with the infection. However, when it comes to pathogen infection, antibiotic usage might bring many adverse effects, such as disturbance to the intestinal microenvironment, allergy and drug resistance. Thus, to develop a dietary intervention strategy plays a pivotal role in confining infection. The results of this research indicate that GOS can effectively protect against the infection of *E. coli* O157 in mice.

Galacto-oligosaccharide (GOS) is composed of 1–7 galactose connected with the galactose or glucose molecule [3]. It is a type of functional oligosaccharide with natural attributions, a small amount of which exists in mammals’ milk, while large quantities exist in human breast milk. Although human digestive enzymes cannot digest GOS, it can be digested by some microorganisms in the intestine and selectively promotes their growth and activity [4]. As a type of prebiotic, GOS has already been added to infant formula and other functional foods to enhance the immune system and promote the digestion and absorption of nutrients [5,6,7,8].

Clinical trials have proved that supplementation of GOS can promote *Bifidobacterium* and other beneficial bacteria, inextricably linked to many diseases’ symptoms. A non-digestible substrate like GOS can promote the growth of Bifidobacterium species [9]. A higher level of *Bifidobacterium* and *Lactobacillus* was found in infants supplemented with GOS, which significantly lowered the presence of colic [10]. The mixture containing GOS was also proved to be effective in fighting *Salmonella* infection [11]. Krumbec et al. found that an amount of GOS intake significantly increased *Bifidobacterium* and lowered gut permeability in a double-blind experiment [12]. Therefore, GOS can inhibit diseases related to intestinal permeability. Gonai et al. also demonstrated that supplementation of GOS ameliorated the decrease of *Bifidobacterium* in diabetics [6].

At the same time, GOS is metabolized into short chain fatty acids (SCFAs) in vivo, and the health-promoting effect of SCFAs is well-known. Takuya has proved that SCFAs can promote epithelial barrier function [13], and lowered pH in the intestinal environment inhibits pathogens’ growth. SCFAs can induce anti-inflammatory responses against infection through various pathways [14]. They can inhibit histone deacetylases (HDACs), activate glucoprotein (GPR) receptor as ligands, and modulate a pro-inflammatory factor and immune reaction through nuclear factor kappa-B (NF-κB) and AMP-activated protein kinase (AMPK) pathways effectively [4]. Therefore, from the aspect of immune cells, SCFAs are also essential for maintaining immune homeostasis. By stimulating colonic sodium and water absorption, SCFAs help to enhance intestinal adaption [15]. It has also been proved that SCFAs help to maintain mucosal function in the intestine, inhibit enterocyte apoptosis, and promote cyto-protection. Supplementation of SCFAs has also been developed as therapy in patients who suffer from short bowel syndrome [16]. Fukuda et al. have proved that *Bifidobacterium* can prevent pathogen infection by producing acetic acid [17]. Studies have demonstrated that an abundance of *Bifidobacterium* is closely related to many chronic diseases such as diabetes, obesity, autism and irritable bowel syndrome [18,19,20,21]. Besides, it is reported that GOS can alleviate diarrhea caused by rotavirus [22], as well as inhibit cholera toxin binding with the GM1 receptor in cells, because the unique structure of GOS binds to pathogens prior to cells in the host [23]. Similarly, colonization of *Salmonella* in poultry and pig can also be prevented [24,25,26]. However, few studies involve the effect of GOS on reducing food-borne infection. We aimed to investigate the preventive effect of GOS against pathogens and provide reference for GOS application in both foods for infants and adults.

## 2. Materials and Methods

### 2.1. Animals and Experimental Design

Six-week-old male BALB/c mice were bought and fed in the Animal Experimental Center of Nankai University. Mice were housed at a certain temperature (25 °C) with a 12 h–12 h dark-light cycle and had free access to a standard diet (Jiangsu Xietong Pharmaceutical Bio-engineering Co., Jiangsu, China) and sterilized water. All the mice were divided into 3 groups, 10 in each group, including the experimental group (GOS group), the model control group (MC group) and the blank control group (CK group). The GOS group received oral administration of 0.2 g GOS/100 g body weight, which took the additive amount of GOS in infant formula as reference. GOS powder was bought from Yuanye Biotechnology Co. (Shanghai, China) and dissolved in phosphate buffer solution. The same volume of physiological saline was given to the other two groups. The process of intragastric administration lasted for 8 weeks, and continued during the 3-week infection period, during which the mice in the MC group and GOS group were challenged with 2 × 10^8^ CFU *E. coli* O157 (ATCC35150) every day. Meanwhile, stools from all the mice were collected by pressing the abdomen gently on alternate days, and then immediately frozen at under −80 °C. Body weight was also recorded. At the end of the 11th week, mice were sacrificed painlessly. The Animal Ethics Committee of Nankai University approved all animal experiments and the ethic code number is SYXK 2019-0001.

Blood samples of mice were carefully collected, and after wiping with 75% alcohol, the abdominal cavities were opened for tissue collecting. As the main infection sites were the ileum and colon, 2 cm-long sections of these two parts were cut down and flushed using PBS. Enteral contents were stored in liquid nitrogen for future analysis of bacteria flora. DNA isolation from bacteria attached to homogenized intestinal tissue was performed taking advantage of a DNA extracting kit following the manufacturer’s protocol. DNA obtained was mixed well with water treated by diethyl pyrocarbonate (DEPC) to adjust to the same concentration. According to the results of real-time PCR, the quantity of *E. coli* O157 can be calculated based on the quantity of bacterial DNA.

### 2.2. Inflammatory Cytokines in Serum and Intestine and Related mRNA Expression

NF-κB signaling involved in inflammation was measured. Levels of tumor necrosis factor (TNF-α), interleukin-6 (IL-6), interleukin-17 (IL-17) and interleukin-1β (IL-1β) in serum and intestine tissue were measured using ELISA kits (NJJCBIO, Nanjing, China) to analyze the inflammatory response caused by *E. coli* O157 infection. Serum samples stored under −80 °C were directly taken for use.

50–100 mg frozen tissue samples were placed into TRIZOL, then homogenized for 5 min. The mRNA within were extracted using TRIZOL kits. In this way, the expression of genes related to inflammatory factors can be quantified with real-time quantitative PCR. The specific PCR primers can be found in supporting information (Appendix A).

### 2.3. Gut Barrier Function Assay

The intestine’s mucous layer plays a marked effect in the gut barrier because pathogens need to penetrate this layer to infect the hosts. Additionally, the amount of tight junction protein is also an important indicator of the gut barrier function [27]. The quantification of gene expression related to gut barrier function (MUC2, ZO1, Claudin and Occludin) in colon tissue was also performed with real-time quantitative PCR.

### 2.4. 16s rDNA Sequencing

Extraction of bacterial DNA from cecal contents, PCR amplicon sequencing and gene analysis were all completed by the Genedenovo Biotechnology Company (Guangzhou, China). 16S rDNA gene sequencing was adopted to analyze intestinal flora. The procedure included extracting microbial DNA and examining gene validity using gel electrophoresis. The specific primers with barcode 341F (CCTACGGGNGGCWGCAG) and 806R (GGACTACHVGGGTATCTAAT) were applied to amplify the V3+V4 domain of 16S rDNA. The amplification was recovered and quantified using a Qubit3.0 fluorimeter, an equal amount of which was mixed and connected to a sequencing joint. According to Illumina’s official details, the sequencing library was constructed, and sequencing was accomplished using the PE250 model in the Hiseq2500 system. After raw data were filtered and combined, the results of operational taxonomic units (OTUs) were obtained when 97% similarity was set as a crosscut. In order to reduce the influence of low-abundance OTU on the whole analysis, OTUs with an abundance under 1 were filtered out without the involvement of subsequent analysis. Based on the species richness information in OTU, the Chao1 index, ACE index, Simpson index and Shannon index were used to get an estimated value of α-diversity. Unweighted pair-group method with arithmetic mean (UPGMA) clustering and principal coordinate analysis (PCoA) were conducted to compare differences between intestinal bacterial communities and forecast their function. Welch’s t-test and linear discriminant analysis effect size) (LEFSE analysis were utilized to learn about indicator species. PICRUSt algorithm was applied to prospect and analyze the function of the microflora metagenome. Canonical correspondence analysis (CCA) and redundancy analysis (RDA) were employed to show the relationship between microbiota and environmental factors, then distance matrix of species and environmental factor were calculated based on Bray-Curtis. A Mantel test was adopted for analyzing environmental factors. To study the correlation between species abundance in various classification hierarchies and different environmental factors, we adopted Pearson correlation analysis and interpreted the environmental factors’ influences on the microbial community structure.

### 2.5. Short-Chain Fatty Acids in Feces

Short-chain fatty acids’ quantification in the fecal sample was performed using gas chromatography. 0.1 g sample was acidified in sulfuric acid and homogenized using 1 mL pure water. After being centrifuged for 5 min at under 10,000× *g*, the clear supernatant extract was easily separated from precipitate. Then the supernatant was mixed well with 0.5 mL ether to let the SCFAs fully dissolve in ether. After 10 min setting time, the liquid was stratified into two phases. The upper organic phase was passed through a nylon membrane with a pore size of 0.22 μm. The determination was performed with gas chromatography equipped with a DB-FFAP capillary column (30 m × 0.25 mm × 0.25 μm, Agilent, Shenzhen, China). Levels of SCFAs were calculated using Agilent’s MSD ChemStation (E.02.00.493).

### 2.6. Statistical Analysis

T-test was adopted to determine the differences between different groups, and Shapiro-Wilk for evaluating the normality of data. Kruskal-Wallis was used for comparing data of non-normal distribution. *P*-value in the nonparametric analysis was adjusted to the Benjamin-Hochberg test from FDR. GraphPad Prism 8.0.2 (GraphPad Software, San Diego, CA, USA) was used for statistical analysis. Significant differences were defined as *p* < 0.05, and the false discovery rate has been adopted to adjust the microbiome/SCFA findings. 16S rRNA genomics data bioinformatic analysis was performed using Omicsmart, a dynamic real-time interactive online platform for data analysis (http://www.omicsmart.com).

## 3. Results

### 3.1. GOS Inhibited E. coli O157 Colonization

Based on previous studies, there are many virulent genes in *E. coli* O157. Meanwhile stx1 was used in this study to determine colonization of *E. coli* in vivo. As shown in Figure 1, the detection level of *E. coli* O157 in the MC group was set as a standard value. In ileum, colonization of *E. coli* O157 in GOS group was only 35% of MC group (Figure 1A, *p* < 0.001) and 38% in the colon (Figure 1B, *p* < 0.001). The number of *E. coli* O157 significantly reduced in the mucous layer.

Pathology slices are shown in Figure 2A. Ileum tissue seemed intact in the CK group, with long villi and deep crypt, and no cellular infiltration could be observed. In the GOS group, the villus height was shorter than that in the CK group; however, the tissue was approximately integral, with partial cell infiltration. Severe damage and inflammatory cell infiltration could be observed in MC group and villus height decreased indicating that intestinal tissue had been damaged. Villus height and villus/crypt (V/C) ratio in the GOS group were significantly higher than in the MC group, while crypt depth was significantly lower than in the MC group (Figure 2B).

### 3.2. Alleviation of Inflammation

With regard to inflammation, infection of *E. coli* O157 often increases inflammation in hosts. ELISA kits were adopted to measure the inflammation level in mice tissue and serum, and the results are shown in Figure 3. In the GOS group, IL-6, IL-1β and TNF-α levels in tissue and serum were significantly lower than the MC group (*p* < 0.01) except IL-17. However, the content of IL-17 in the GOS group showed no significant difference with the MC group.

The expression of inflammatory cytokines also showed a similar trend (Figure 4). Compared with the CK group, IL-6, IL-1β, TNF-α and IL-17 all increased in the MC group under the infection of *E. coli* O157, while the GOS group reversed this increase under intervention. Levels of IL-6 and IL-1β can reflect inflammation in the body directly. IL-17 was secreted by CD4+T cells, while CD4+T cells promoted the relieving of proinflammatory factors. TNF-α can mediate the activation of neutrophilic and so mediated inflammatory reaction. Inflammatory response emerged in mice when infected by *E. coli* O157 or interfered with by the toxin. The rise of IL-17 and TNF-α suggested the activation of Th17 immunocyte, while IL-1β and IL-6 indicated humoral immunity activation.

### 3.3. Enhancement of Gut Barrier Function

The intestinal mucous layer consists of mucin. The expression of mucin and tight junction protein reflected the health condition of the gut barrier directly. Real-time fluorescent quantitative PCR was adopted to detect the relative expression level of MUC2, ZO1, Claudin and Occlaudin in colon tissue, and the results are shown in Figure 5. Expression of MUC2 and Occlaudin were significantly downregulated (*p* < 0.05) in the MC group compared to the CK group, which suggested that infection of *E. coli* O157 suppressed the expression of part of the protein related to gut barrier function. The expressions of various proteins related to this function all increased with respect to the GOS group, compared to the MC group (*p* < 0.05). In particular, the expression of Occlaudin was 2.8 and 4.5 fold higher than in the CK group and the MC group, respectively.

### 3.4. Increase of SCFA Contents

Content of acetic acid, propionic acid, butyric acid and valeric acid were all quantified and the results are shown in Figure 6. All types of SCFA in the MC group were significantly lower than the CK group (*p* < 0.001), among which the concentration of propionic acid and butyric acid were only half that of the CK group. With regard to the GOS group, levels of acetic acid, propionic acid and valeric acid showed no significant differences from the CK group except for butyric acid. The butyric acid level in the GOS group was still significantly higher than the MC group (*p* < 0.05). These results demonstrated a protective effect of GOS against the decrease of SCFAs caused by *E. coli* O157 infection. Concentration of SCFAs in feces of each group can be found in Appendix A.

### 3.5. Effect of GOS on Gut Microbiota Composition

The microbiota in the cecum of mice was sequenced to study the effect of GOS on the intestinal microbiota of mice infected by *E. coli* O157. From the distribution of species abundance, Firmicutes and Bacteroidetes were the dominant species at the phylum level (Figure 7A). Compared with the CK group, the abundance of Firmicutes and Bacteroidetes in the MC group decreased, while Proteobacteria increased substantially. The relative abundance of Firmicutes decreased by GOS relative to the CK group, and Proteobacteria abundance rose. Since *E. coli* O157 belongs to Proteobacteria, it is easy to understand the increase of Proteobacteria. Simultaneously, the abundance of Bacteroides also increased by GOS intake relative to the MC group. Utilizing the linear discriminant analysis effect size (LEfSe) analysis, differences between the bacterial genera abundance of mice in different groups were identified. At the genus level (Figure 7B), Escherichia_Shigella, Akkermansia_muciniphila and Ruminococcaceae_UCG_010 were significantly more abundant in the GOS group than in the CK group (Figure 7C), while Vibrio, Ruminococcaceae_UCG_010, Akkermansia_muciniphila and Bacteroides_vulgatus were significantly more abundant in the MC group. The abundance of Bilophila, Enterorhabdus_muris, Candidatus_Saccharimonas and Patescibacteria were also downregulated in the GOS group compared to the CK group (Figure 7D), which was not observed in the MC group (Figure 7E). This may be due to the regulatory effect of GOS on the flora.

In terms of species alpha diversity, the GOS group was slightly lower than the MC group, and the MC group was slightly lower than the CK group (Figure 8A). From the sequencing results of principal coordinate analysis (PCoA) (Figure 8B), the first principal coordinate is 23.85%, and the second principal coordinate is 21.38%, with good interpretability. From the point of view of distribution, the CK group was clustered, which was significantly different from the MC group and the GOS group, indicating that high intake of *E. coli* O157 had a significant impact on the intestinal flora diversity of mice. PICRUSt predicted the gut community metagenome of the mice. Heat map results (Figure 8C) showed that the level of glycan biosynthesis and metabolism of the MC group was significantly lower than that of the CK group (*p* < 0.05), while the incidence of infectious diseases was significantly higher than that of the CK group (*p* < 0.01). The abilities of terpenoids and polyketides in terms of their metabolizing, replication and repairing, translating, folding, sorting and degrading in the GOS group were significantly lower than those in the MC group (*p* < 0.01). At the LV3 (Figure 8D), the abilities related to pantothenate and COA biosynthesis, aminoacyl tRNA biosynthesis, fatty acid biosynthesis, D-alanine, metabolism, protein export, terpenoid backbone biosynthesis, glycolysis gluconeogenesis, pyruvate metabolism, nicotinate and nicotinamide metabolism, base exception repair, purine metabolism, nucleotide exercise repair, taurine and hydroxy-tryptamine metabolism, and RNA degradation in the GOS group were significantly lower than those in the MC group (*p* < 0.01). At the same time, glycosaminoglycan degeneration was significantly higher (*p* < 0.01).

In order to study the correlation between intestinal microflora and other sample indicators, we introduced environmental factor analysis and established the CCA model of microbial community samples along with four main factors, including the total amount of SCFAs, the relative expression of MUC2 and Occludin, and the relative expression of IL-6 in the colon. The results are shown in Figure 9A. CCA1 and CCA2 accounted for 61.52% and 26.92% of the whole microbial community, respectively, and the overall level appeared relatively high. SCFAs and Occludin were significantly correlated with the distribution of microbial communities (*p* < 0.05). In terms of the correlation between species, Figure 9B showed that *Akkermansia*, *Ruminococacaeae_ Ucg-010*, *Muribaculum* and *Bacteroides* were all positively correlated with Occludin expression, while *Lachnospiraceae*, *Eubacterium_ ruminantium_* Group and *Eubacterium_ xylanophilum_* Group were positively correlated with IL-6 and negatively correlated with MUC2.

## 4. Discussion

As prebiotics, it is widely perceived that GOS can modulate gastrointestinal microflora and bring benefits to hosts [28]; however, the mechanism has not been reported. After oral intervention for three months, the adhesion of *E. coli* O157 was significantly decreased by 65.0% and 62.2% in the ileum and colon mucus layer. GOS protected intestinal villi integrity, reduced inflammation in blood and tissue, and improved SCFA levels in hosts [27,29]. Moreover, we also found that the intake of GOS significantly upregulated the expression of mucin in the intestine, which might be attributable to its regulation to specific microbiological species. It is also the primary approach for GOS to prevent hosts from pathogen invasion.

Enterohemorrhagic *E. coli* O157 mainly enters the intestine through the oral route, then passes the mucus layer and reaches epithelial cells. *E. coli* O157 can bind with a specific immuno-determinant on epithelial cells and release toxins to damage the cells. The results of animal experiments demonstrated that GOS could significantly reduce the number of *E. coli* O157 reaching the mucus layer, thus protecting the host from infection. A similar result was also reported in Ignasi’s research. It was found that GOS provided significant resistance to rotavirus infection and alleviated diarrhea [22]. This was relevant to the unique structure of glycoconjugates in GOS, which can restrain viruses from binding with hosts and thus protect hosts from infection. Results of 16s DNA sequencing showed no difference in the relative abundance of *E. coli* between MC and GOS group. No pathogen reduction in the intestine indicated that GOS itself might not have bacteriostasis; instead, it prevented *E. coli* O157 from reaching the enterocyte through the mucus layer. We speculate that the result should be linked with the modulation of the gut barrier function.

In previous studies, Wang et al. have demonstrated that GOS can upregulate gene expression related to barrier function (ZO-1 and Claudin-1) [30,31]. Perdijk also proved GOS modulated epithelial cell functioning in vitro [32]. Herein, up-regulated expression of tight junction proteins, including Occludin, Claudin and ZO1, as well as mucus protein MUC2 in intestinal tissue, demonstrated the effect of GOS in vivo. Particularly, Occludin expression was 1.6-fold induced with GOS compared with the CK group, and 4-fold compared with the MC group (Figure 5). As a protein performing barrier function, Occludin plays an important part in regulating intestinal permeability. On the other hand, MUC2 is a mucus protein that avoids intestinal epithelium exposure to pathogens. In the present research, after invading and injuring the enterocyte, *E. coli* O157 down-regulated the expression of mucin and tight-junctional protein, which caused thinner mucous layer and bigger intercellular space and permeability. As a result, more enterotoxin and pathogens travelled from the mucous layer to the enterocyte and even entered the bloodstream. All these processes were caught in a vicious circle and finally induced systemic inflammatory response. The intervention of GOS enhanced the expression of mucin and tight junction protein, therefore preventing the pathogens’ first invasion. The damage circle was retarded and alleviated, and the hosts were protected. We infer that the modulation of intestinal flora should induce these effects.

The beneficial and robust modulation of GOS on gut microbiota could be verified and partly interpreted by the sequencing result of cecal contents. For example, at the genus level, the abundance of *Akkermansia_muciniphila* and *Ruminococcaceae_UCG_010,* which can produce SCFAs effectively, improved significantly in the GOS group. The abundance of *Bacteroides_vulgatus,* known as symbiotic bacteria preventing *Vibrio cholerae* infection, also rose significantly [33]. The increase of these beneficial microorganisms indicated that GOS could promote intestinal flora health even under a large amount of *E. coli* O157 gavage. Similar results for GOS can be found in previous studies. Interestingly, *Vibrion*, previously known as harmful, also increased significantly in the GOS group. Constrained by sequencing depth, we do not know the exact abundance of which species in *Vibrion* improved. PICRUSt predictive analysis showed no significant differences in rating scores of disease-causing floras between GOS and MC group. Only an upregulation of glycosaminoglycan degeneration function indicated that the increase of the *Vibrion* genus might be caused by the increase of species that can metabolize glycan independently. With an abundance of less than 0.003% in the GOS group and 0.0003% in the MC group, *Vibrion*’s abundance was seen only minimally from the OTU result. Hence, they would not make a difference to the flora function. We should still pay continuous attention to this. Moreover, several studies have revealed the promotion effect of GOS on *Bifidobacterium*; however, this was not detected in our study, which might be caused by differences in animal models. Models that are closer to the human intestinal environment should be utilized in future research. In this research, GOS played a role in reducing inflammation and enhancing gut health, and its role did not depend on the abundance of *Bifidobacterium*. This result was rarely reported in previous studies. Because the abundance of *Firmicutes* was negatively correlated with the pathogen population, and the *Bacteroidetes* population was greatly influenced by infection, the *Firmicutes/Bacteroidetes* ratio could be seen as an indicator of inflammation [34]. The CK group ratio was 1.59, which was the lowest in the three groups, while the MC group ratio was the highest and the GOS group was in the middle. This ratio relationship showed that GOS alleviated inflammation caused by *E. coli* O157 to some extent.

SCFAs are a type of well-known, beneficial organic acid, among which acetic acid can modulate pH in the intestine and create an acidic environment to inhibit pernicious bacteria [35]. Propionic acid can promote the expression of human leukocyte antigen and immunoglobulin, and butyric acid can be absorbed and utilized by epithelial cells as energy [36,37]. As an influential signal factor in the brain-gut axis, pentanoate can modulate intestinal motility [38]. Production of SCFAs mainly come from the fermentation of beneficial microorganisms such as Bacteroides and Firmicutes [39]. Many studies have suggested that prebiotics such as beta-glucan and inulin improve SCFA level in the gut [40,41]. In this research, the invasion of *E. coli* O157 caused damage to the gut environment and interrupted the balance among gut flora. The significantly decreased SCFAs were held back by GOS intake. Two aspects should be accountable for this phenomenon. On the one hand, GOS intake lowered the colonization number of *E. coli* O157 in the intestinal mucous layer and defended against the infection and damage to intestinal tissue. As long as mucin is expressed in goblet cells, flora like *Akkermansia_muciniphila* living on mucin can normally metabolize producing SCFAs [42]. Because of *E. coli* O157 infection in the MC group, the intestinal epithelium was seriously damaged, and mucin production was down-regulated; thus, SCFAs produced by bacteria living on mucin decreased sharply. On the other hand, with low molecular weight, it is widely reported that GOS can be utilized by colonic microflora and metabolized into SCFAs (acetic acid, propionic acid, butyric acid and pentanoate), lactic acid, succinic acid, gas (CO_2_, CH_4_, H_2_ etc.), as well as a small amount of formate, acetate, valerate, 2-Methyl butyrate and isovalerate [43]. The contents of SCFAs in mice feces were also measured before infection, and the results were similar (Appendix A). Studies have shown that SCFAs can enter cells to help maintain immune homeostasis through blocking HDAC regulation of the NF-κB pathway [44,45,46]. Proinflammatory cytokine production, such as TNF-α, IFN-γ and interleukin, was inhibited [47]. In this research, alleviation of the inflammatory level might be a joint result of the reduction of *E. coli* O157 colonization in the mucous layer and the increase of SCFAs.

According to Mack in earlier research, probiotics can inhibit *E. coli* adhesion in vitro through inducing expressions of mucin gene MUC2 and MUC3 [48]. Wang et al. also suggested that increasing SCFAs promoted the expression of MUC2 [30]. Therefore, we assumed that GOS might up-regulate mucin and tight junction protein expressions by promoting the bacteria that produced SCFAs. Our correlation analysis of environmental factors and gut microflora suggested that the intake of GOS increased the species richness of *Akkermansia_muciniphila* and *Ruminococcaceae_UCG_010*, which were significantly correlated with the expression of Occludin. At the same time, SCFA-producing flora like *Muribaculum* and *Bacteroides* showed similar results. It is noticeable that they were also positively correlated with SCFA content, however, not significantly. This result indicated that species that were extremely correlated with Occludin expression participated in feedback regulation through other unknown signal pathways (for instance, lipopolysaccharide) than SCFAs. Dietary intervention by GOS irrigation promoted *Akkermansia_muciniphila* and other probiotics, which contributed to the homeostasis of intestinal flora. Through SCFAs and/or other pathways, these intestinal florae stimulated goblets and epithelial cells to produce more mucin and tight junction protein to enhance gut barrier function. Thus, less *E. coli* O157 was able to pass the mucous layer and reach the epithelial cells. Moreover, the significant positive correlation between *E. coli* abundance and Occludin expression might be due to the low-toxic and long-term pathogen adopted in this research which may be due to the protective feedback regulation of intestinal epithelial cells caused by the long-term infection of low virulent *E. coli* O157, thus up-regulating the expression of Occludin. In addition, the *Lachnospiraceae*, *Eucharaterium_ xylanophilum_* Group and the *Eucharaterium_ ruminantium_* Group had significantly negative correlations with the expression of IL-6. Few studies have researched these florae, and further investigation can be done to see whether these can become new biomarkers for flora disturbance.

In our present study, we discovered the biomarkers that played protective roles in GOS protection against *E. coli* O157 infection. Limited by experimental conditions, independent studies on *Akkermansia_muciniphila* and *Bacteroides_vulgatus* in vivo or in vitro have not been carried out. Follow-up studies on various pathogens, GOS interventions under different concentration gradients, and animal models closer to the human intestinal environment are still needed.

## 5. Conclusions

Overall, we have proved that GOS can reduce pathogens’ colonization of the intestinal epithelium by enhancing gut barrier function. GOS significantly promoted the abundance of *Akkermansia_muciniphila*, *Ruminococcaceae_UCG_010* and *Bacteroides_vulgatus*, thus increasing SCFA levels in the intestine. Expressions of mucin and tight junction protein were also up-regulated. Recent research has suggested that *Akkermansia spp* might be associated with the treatment of chronic illnesses, such as obesity and diabetes, and weak agonistic LPS of *Bacteroides_vulgatus* restores intestinal immune homeostasis. Therefore, enhancing these beneficial bacteria selectively probably provides a new means to prevent and treat these diseases. Our research revealed the mechanism behind GOS protection against pathogen infection and provided a theoretical basis for modulation of gut health.

## Figures and Tables

**Figure 1 foods-09-01710-f001:**
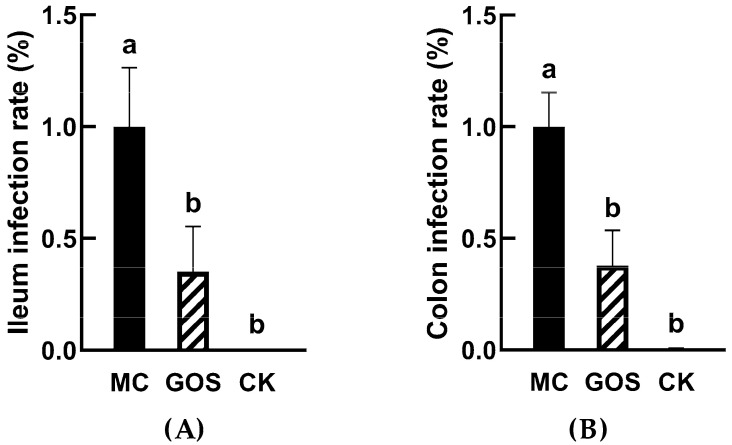
*E. coli* O157 colonization in ileum and colon (**A**) Colonization in ileum; (**B**) Colonization in colon. Different letters (a,b) showed significant difference (*p* < 0.05).

**Figure 2 foods-09-01710-f002:**
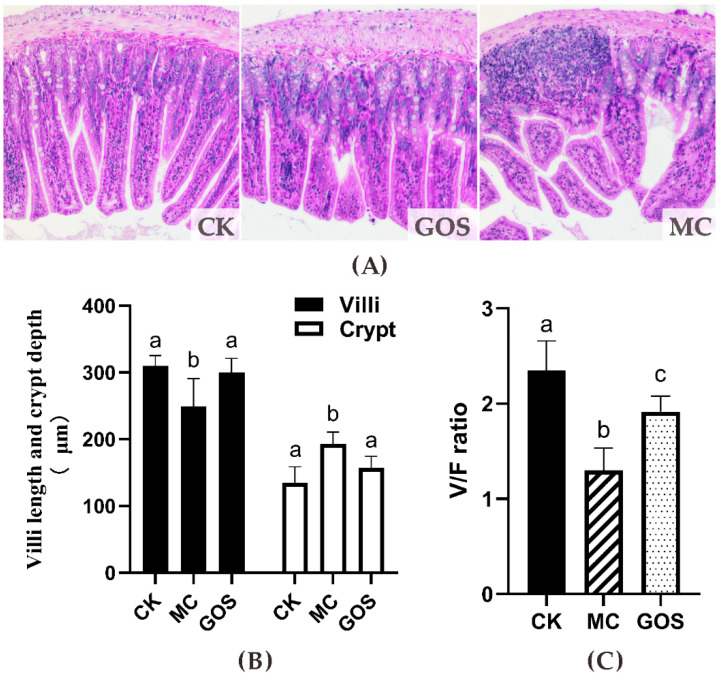
Effect on the height of the villi and the depth of the fossae in the small intestine. (**A**) Photomicrographs in the small intestine of the mice (HE×100); (**B**) Height of the villi (V, μm), depth of the crypts (F, μm); (**C**) the V/F ratio. Different letters (a–c) show significant difference (*p* < 0.05) in the same tissues among groups.

**Figure 3 foods-09-01710-f003:**
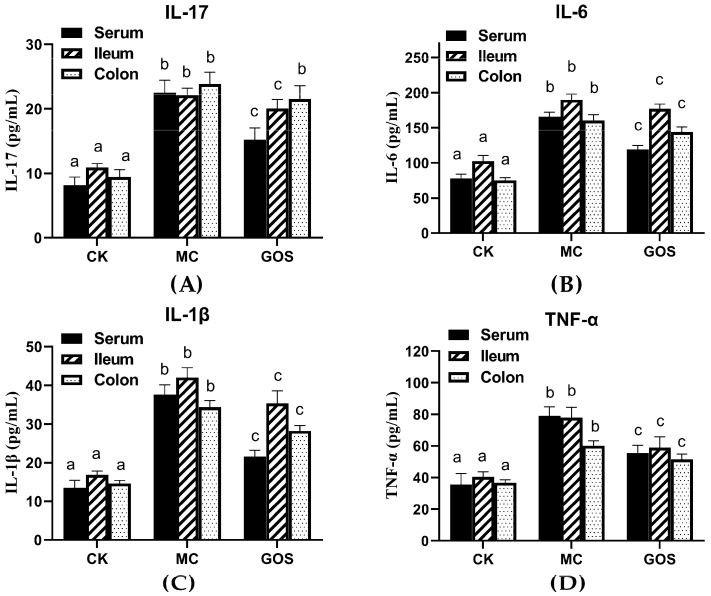
The concentration of inflammatory factors in different tissues. (**A**) The concentration of IL-17; (**B**) The concentration of IL-6; (**C**) The concentration of IL-1β; (**D**) The concentration of TNF-α. Different letters (a–c) showed significant differences (*p* < 0.05) in the same tissues among groups.

**Figure 4 foods-09-01710-f004:**
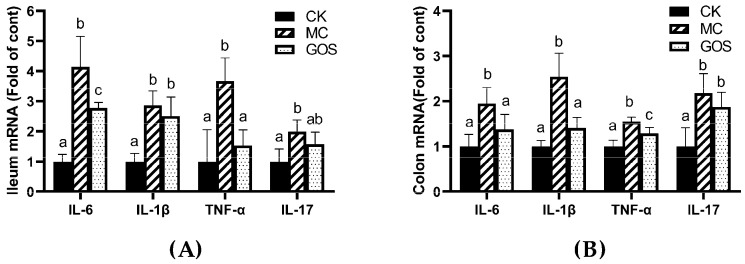
Relative mRNA levels of inflammatory factors, all normalized to β-actin mRNA expression. (**A**) Inflammatory factors in ileum; (**B**) Inflammatory factors in the colon. Different letters (a–c) are used to express the significant differences (*p* < 0.05) of the same factor among groups.

**Figure 5 foods-09-01710-f005:**
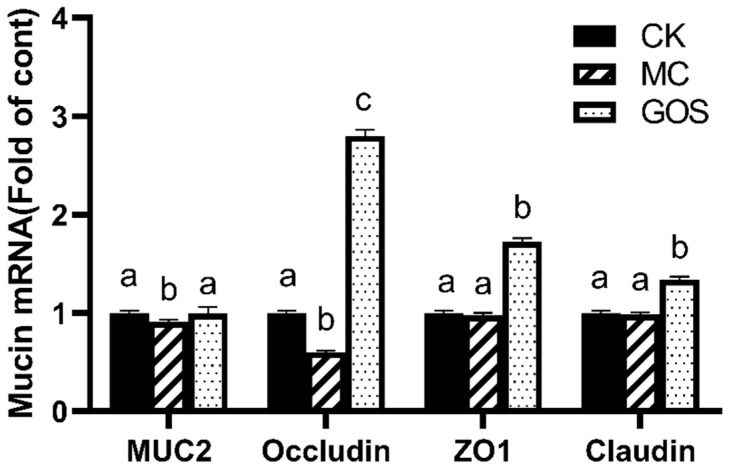
Relative mRNA levels of mucins, all normalized to β-actin mRNA expression. Different letters (a–c) are used to express the significant difference (*p* < 0.05) of the same factor among groups.

**Figure 6 foods-09-01710-f006:**
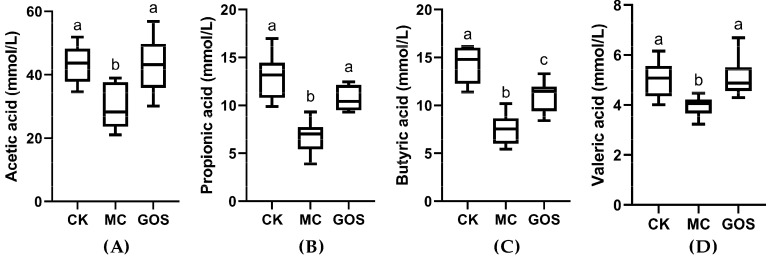
Concentrations of various short-chain fatty acids (SCFAs) in feces after infection. (**A**) Acetic acid; (**B**) Propionic acid; (**C**) Butyric acid; (**D**) Valeric acid. Different letters (a–c) are used to express the significant differences (*p* < 0.05) between various short chain fatty acids (SCFAs).

**Figure 7 foods-09-01710-f007:**
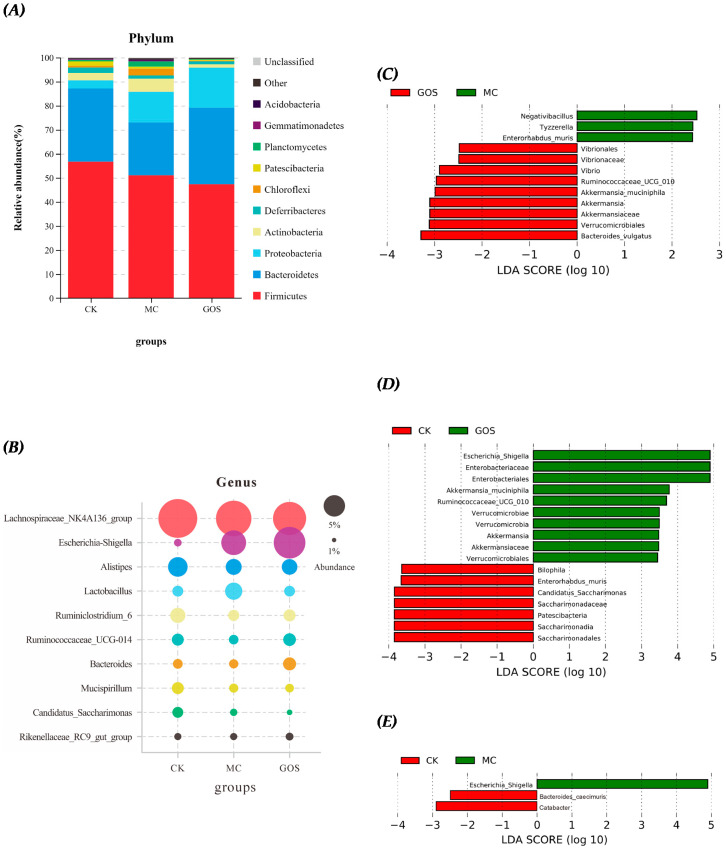
Relative abundance and Linear Discriminant Analysis (LDA) of intestinal flora in different groups of mice. (**A**) Relative abundance of phylum level; (**B**) Relative abundance of genus level; (**C**) LDA in MC group and galactooligosaccharide (GOS) group; (**D**) LDA in blank control (CK) group and GOS group; (**E**) LDA in model control (MC) group and CK group.

**Figure 8 foods-09-01710-f008:**
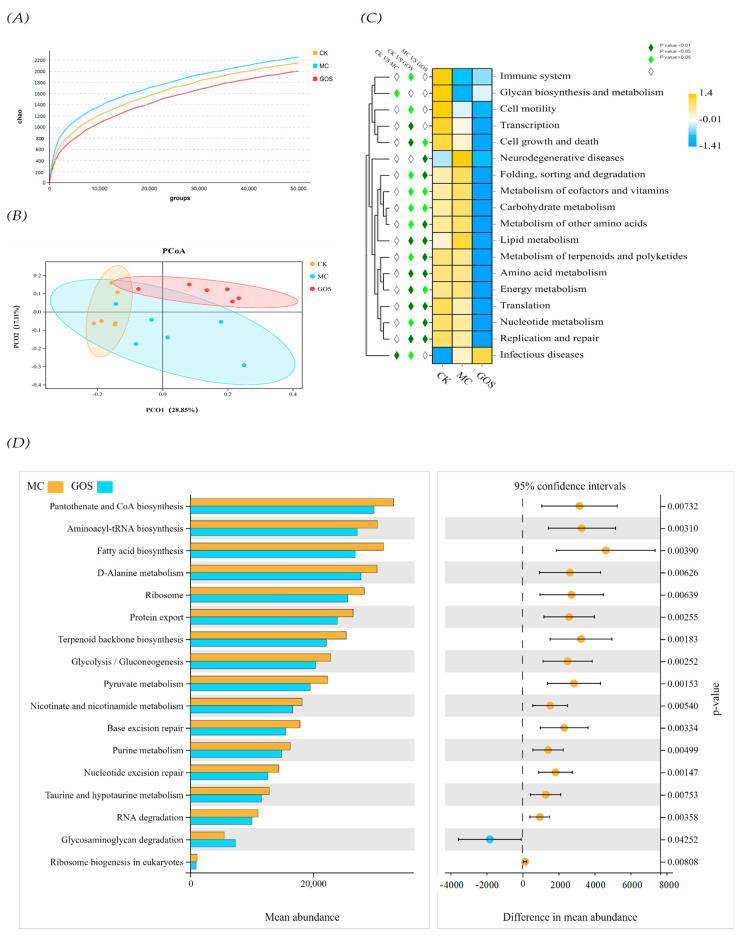
The diversity and function prediction of intestinal flora in different groups of mice. (**A**) Dilution curve; (**B**) Principal Co-ordinate Analysis (PCoA) based on operational taxonomic units (out) results; (**C**) Heat map of function prediction; (**D**) Mean abundance of differential functions.

**Figure 9 foods-09-01710-f009:**
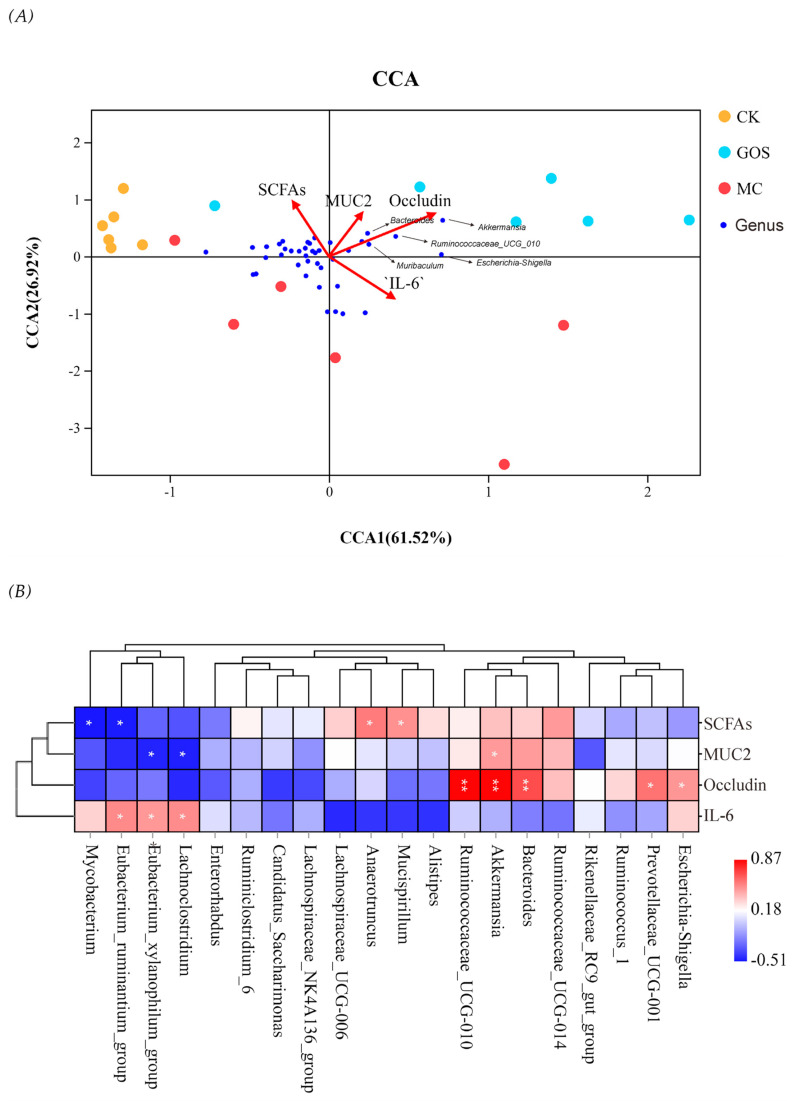
Correlation analysis of environmental factors. (**A**) Canonical correspondence analysis (CCA) of microbial population distribution and environmental factors. Environmental factors are marked by red arrows. The angle between the arrow lines and the sorting axis indicates the correlation between environmental factors and the sorting axis. Environmental factors with longer arrow line had a higher correlation with the distribution of intestinal flora. (**B**) As shown in the Spearman correlation thermogram, there were significant differences in bacteria abundance between different environmental factors. The colors range from blue (negative correlation) to red (positive correlation). The significant correlation was expressed by * *p* < 0.05 and ** *p* < 0.01.

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
