# Peer review of "Protection of Galacto-Oligosaccharide against E. coli O157 Colonization through Enhancing Gut Barrier Function and Modulating Gut Microbiota"

_foods, 2020, doi:10.3390/foods9111710_

Round 1

Reviewer 1 Report

Authors in presented article entitled "Protection of Galacto-oligosaccharide against E.coli O157 colonization through enhancing gut barrier function and modulating gut microbiota" decided to investigate correlations between GOS and intestinal homeostasis in presence of E. coli infection. Research articles concerning modification of gut microbiota and its impact on host's wellbeing are a so called "hot topic" in scientific community and manuscript presented by authors is in this area of research. 

Broad comments

The design of the study is proper and results are interesting and promising. Many research techniques were included in analysis of changes in microbiota composition production of its metabolites, changes in concentrations of inflammatory cytokines and mRNA for gut-blood barrier proteins what is an asset of this study. However there are a few concerns that diminish value of presented manuscript. This article requires extensive and profound English language and style correction. There are plenty of mistakes and some parts are not understandable at all. Additonally introduction and methodology sections should be revised together with improving figures. 

Specific comments

  1. The number of language errors exceeds my ability to list them in this section. I can write a few examples, but the whole manuscript should be spell checked. 
    line 30-31 - the sense of this sentence is like meat and diary would cause diarrhea and other symptoms.
    line 39 after although "it" is missing
    line 43 i would change nutrition to nutrients (just a suggestion)
    line 53 - large amount of studies can be changed to "Many"
    line 64 - had free access
    line 85 - not is -> was
    line 88 - samples were not was
    line 94 - plays not play
    line 124 and 347 - feces of fecal content, because fecal is an adjective 
    line 144-145 i would reccomend changing sequence of words to make it more clear for readers.
    line 154 - insted became less i would recommend "decreased"
    line 191 - proteins
    line 198 - concentrations insted of contents
    line 245 - using word obvious is rather colloquial 
    Discussion section also includes many mistakes and should be corrected.

  2. There are some not clear statements in introduction 
    line 44-45 I don't see connection between GOS Biffidobacterium and symptoms of diseases. It is not clear and should be explained and revised.

  3.  In introduction i would recommend extending paragraph on positive effects of SCFA on gut physiology with including more references.
  4. In line 89 in methodology section authors wrote that homogenization lasted for long enough. It is not professional to write like that. Methodology description should be more precise. It is not specific what does it mean long enough?

  5. In figure 4 there is graph C that has not been described in title of figure and additonally the same graph is on Figure 5. Why is that?

  6.  In figure 5 authors wrote that concentrations presented on figure were assesed before and after infection. However the results on figure looks like from the same time point between different groups. It is a serious flaw, because it is not clear whether samples were collected only once or twice and also differenct statistic tests would be needed to assess those changes. Moreover group CK was not infected at all, so what should be written there and how should be analyze this graph?
  7. In figure 7 graph C is difficult to find i would reccomend changing layout to make it more visable, because at first I could not find it. 

Reviewer 2 Report

The authors Zou et al. have investigated the role of galacto-oligosaccharide against E.coli infection and looked at their gut microbiota and barrier integrity. To do so, they used they 30 Balbc mice, divided into three groups, of which, to one group was infected for 8 weeks and collected intestinal tissue samples, performed 16s gene seq and targeted metabolomics for SCFA metabolites, mRNA tissue for inflammatory markers. Overall, they have found beneficial effects of GOS post-infection with E.coli such positive modulation of microbiome, immuno-modulatory effects, and SCFA production. Essentially, GOS acted a the potential probiotics during infections. However, the authors needs to address the following comments before publication.

Comments:

  1. The authors haven't mentioned anything about the disease activity index? What about the body weigth changes, overall health changes during infection? I suggest authors to provide this information.
  2. Did the authors collected feed-related information? what was the diet? please provide the detail
  3. Methods 2.4 16s gene seq, line 100, authors mentioned they sequenced cecal contents, however, they mentioned feces in results, line 200, 3.5 results. Assuming the authors have collected fecal, cecal contents, it will a wonderful opportunity for the authors to include and compare fecal vs. cecal microbiome.
  4. Whether the SCFA was from fecal or cecal contents, again it will a wonderful opportunity for the authors to include and compare fecal vs. cecal SCFA.
  5. Did the authors correct their microbiome/SCFA findings by utilizing false discovery rate ?

Strengths and limitations: The biggest strengths of the work are use of adequate controls and number of animals per group. The limitations are use only male animals, as emerging research demonstrates role of sex hormones in infection and microbiota.

Minor comments:

Use of a figure/illustration for methods could be beneficial.

0.2 g GOS/100 g, how did the authors determine the dosing. Have they performed any invitro studies to determine this?

Figure 9, heatmap, there is a column "SCFAs", which SCFA are the authors talking about? How did they quantify all SCFAS together?

I suggests authors to look at firmcutes/Bacteroidetes ratio and include it in the discussion. It stands as an indicator of inflammation.

Did

Round 2

Reviewer 1 Report

Authors revised manuscript entitled: "Protection of Galacto-oligosaccharide against E.coli O157 colonization through enhancing gut barrier function and modulating gut microbiota" and uploaded corrected version for second review. 

Authors implemented reviewer's suggestions in corrected version of the manuscript. The research interest and quality of presented results significantly improved. Introduction section has been extented and includes interesting new facts on positive impact of SCFA on gut health. 

New figures made presented data easier to read and analyze, and are more interesting for the readers. Authors additonally removed methodological errors in figure legends in present version of the article. 

I believe that overall quality and significance of the article improved in present version. The research topic is very interesting and will attract researchers investigating correlations between gut microbiota and host's wellbeing.  

There is still some language correction needed:
line 65: lowered the presence of colic presence - what authors meant by that?

Reviewer 2 Report

The authors have sufficiently revised the manuscript.

Author Response

Dear Editors and Reviewers:

Thank you very much for your letter and comments on our manuscript entitled “Protection of Galacto-oligosaccharide against E.coli O157 colonization through enhancing gut barrier function and modulating gut microbiota” (ID: Foods-976168). We greatly appreciate the constructive comments that are very helpful for our revision of the manuscript. We will continue to work hard in our future research. Wish you the best of health and success!